# Emotional State of Young Men in Relation to Problematic Internet Use

**DOI:** 10.3390/ijerph191912153

**Published:** 2022-09-26

**Authors:** Natalia Tomska, Aleksandra Rył, Agnieszka Turoń-Skrzypińska, Aleksandra Szylińska, Julia Marcinkowska, Damian Durys, Iwona Rotter

**Affiliations:** 1Department of Medical Rehabilitation and Clinical Rehabilitation, Pomeranian Medical University, Żołnierska 54b, 71-210 Szczecin, Poland; 2Clinical Department of Nephrology, Transplantology and Internal Medicine, Pomeranian Medical University, Al. Powstańców Wielkopolskich 72, 70-111 Szczecin, Poland; 3Student Science Club “KINEZIS”, Department of Medical Rehabilitation and Clinical Physiotherapy, Pomeranian Medical University, Żołnierska 54b, 71-210 Szczecin, Poland

**Keywords:** Internet, behavioral addiction, young man, male

## Abstract

The Internet has become an indispensable tool in communication, business, entertainment, and obtaining information. Behavioral addictions are disorders associated with uncontrolled activity feeding the reward system, motivation, and memory. The purpose of this study was to assess the emotional state in terms of problematic Internet use. The survey was conducted in 2020–2021 in the West Pomeranian region of Poland and involved 500 men aged 18–30 (24.82 ± 3.83). The study was conducted using our own original questionnaire regarding the amount of time spent playing computer games during the weekdays and on days off; the type of school/university; financial situation; as well as the manner, purpose, and degree of Internet use. Other questionnaires were also used, i.e., Beck Depression Inventory, Internet Use Test, GAD-7 Generalized Anxiety Assessment Questionnaire, and Buss and Perry Aggression Questionnaire. Problematic use of the Internet may result in the occurrence of anxiety, anger, hostility or depression. Longer duration of Internet activity was correlated to higher scores on the Internet Use Test degree of problematic Internet use. There was a correlation between the severity of mild depression symptoms and the occurrence of anxiety, verbal and physical aggression, and problematic Internet use.

## 1. Introduction

The Internet has become an integral part of the daily lives of adults and young people all over the world, as well as an indispensable tool in communication, business, entertainment, and obtaining information [1]. Over the past 15 years, the number of Internet users has increased by more than 1000% [2], along with a similar proliferation of literature on Internet addiction (IA). While the problem of IA is not yet well understood, research into its etiology and course is also still in its early stages [3].

Behavioral addictions are disorders associated with uncontrolled activity [4] feeding the reward system, motivation, and memory [5]. These behaviors can be seen in playing video games, gambling, shopping, sunbathing, sexual activity, work, and others [4]. At least two biological factors contribute to the occurrence of behavioral disorders, namely malfunctioning of the serotonergic and dopaminergic systems [6]. Researchers increasingly demonstrate the relationship of neuromodulators such as dopamine (DA) and serotonin (5-hydroxytryptamine; 5-HT) on the aspects of adaptive behavior, motivation, reward processing, reinforcement learning, and behavioral flexibility. Previously, it was believed that DA and 5-HT are antagonists, but currently, we know that 5-HT promotes reward seeking through the activation of DA neurons [7]. Low levels of 5-HT can lead to the development of anxiety disorders, depression, and addictive behaviors [8]. 

Internet addiction is a new psycho-behavioral disorder seen as repeated or excessive use of the Internet, with a strong and irresistible desire to use the Internet as often as possible, leading to considerable negative social, psychological, and health issues [9]. Withdrawal reactions can cause psychological and somatic withdrawal symptoms [10]. IA is also associated with sleep delays where, in order to have more time for entertainment or relaxation, Internet users tend to postpone their bedtime, which is consistent with the uses and gratifications theory [11]. In other words, satisfaction and happiness are experienced when using technology, but the degree of reward decreases with repetition and increased time of Internet use. Of course, IA, like many other types of addiction, may cause many negative effects not only in terms of sleep neglect, but can also result in inappropriate behavior, somatic and psychosocial problems, hostility and aggression, as well as depression, anxiety, and stress [12]. 

The definition of problematic Internet use is a subject of regular discussion in scientific circles. In general, despite the great interest of researchers in Internet addiction, this area still lacks consistent terminology. For example, terms such as ‘problematic Internet use’, ‘Internet abuse’, ‘maladaptive Internet use’, ‘compulsive Internet-related behavior’ or even ‘Internet addiction’ are being used interchangeably [10]. Researchers can agree that lengthy Internet use may cause stress, consume a large amount of time, and impair functioning in important areas of life [13].

It has been shown that males aged 15–25 are more likely to become addicted to the Internet or online games—3.7% of all Polish males meet the given criteria for gambling addiction, and a further 5.3% are at risk of addiction [14]. Polish male gamblers are aged 30–39, and women 40–49. Symptoms of addiction affect 21.4% of male gamblers and 9.1% of the women. Although there are many positives associated with the Internet and its accessibility, it has an increasingly negative impact on the lives of young people. Therefore, it is important to recognize the clinical signs and symptoms of Internet addiction, treat comorbidities, and initiate psychosocial interventions [15]. 

Researchers have long studied the relationship between addiction, its causes, and negative effects. There is a question of whether negative symptoms exacerbate the onset of addiction, or whether the addiction exacerbates the onset of symptoms. Some point to evidence that inappropriate Internet use can cause depression and anxiety [16,17], while others evidence the reverse, that depression and anxiety can cause addiction [18,19], and others can demonstrate a mutual relationship between addiction, depression, and anxiety [20].

Inappropriate use of the Internet, online games, and extreme sports (as well as stimulants) have a huge impact on people’s lives. The irresistible urge to check online content leads to a lack of control over behavior, which poses the danger of developing behaviors negatively affecting the health of an individual and society [21]. Several reports highlight the serious negative consequences of problematic Internet use—depression, impulsiveness, aggression, anxiety, and social avoidance. The relationship between psychopathological symptoms and behavioral disorders such as addiction is complex and bidirectional [22].

The purpose of this study was to assess the emotional state of young men, such as the symptoms of depression, anxiety, aggression, anger, and hostility, in terms of problematic Internet use.

## 2. Materials and Methods

### 2.1. Survey Sample

The study was a cross-sectional study. The survey was conducted in 2020–2021 in the West Pomeranian region of Poland and involved 500 men aged 18–30 (24.82 ± 3.83).

Inclusion criteria for the study were age (18–30), declared Internet use, playing computer games and/or online games including gambling, and a fully completed questionnaire. Exclusion criteria were being under the care of a psychiatrist or psychologist, and the presence of oncological or endocrine diseases. Participation in the study was voluntary and anonymous. Recruitment for the study was done through online forums, advertisements at healthcare centers, and social media.

### 2.2. Analysis of Risky Behaviors Reported in the Questionnaires

The study analyzed the results obtained from the questionnaires and the degree of risky behaviors in terms of Internet use. In the study group, the 458 men had no chronic diseases, 13 of whom had asthma and some suffered from type 1 diabetes, hypothyroidism, migraines, allergies, atopic dermatitis, hearing disorders, and other conditions, but which were not criteria for exclusion from the study. In the study, 89.8% (n = 449) of the men lived in urban areas, 10.2% (51%) in rural areas; 38.2% (n = 191) of the men rated their material situation as very good, 59.4% (n = 297) as sufficient, and 2.4% (n = 12) as bad.

### 2.3. Questionnaires

The study was conducted using our own original questionnaire regarding the amount of time spent playing computer games during the weekdays and on days off; the type of school/university; financial situation; as well as the manner, purpose, and degree of Internet use. Other questionnaires were also used, i.e., Beck Depression Inventory, Internet Use Test, GAD-7 Generalized Anxiety Assessment Questionnaire, and Buss and Perry Aggression Questionnaire.

Beck Depression Inventory is a tool used in the diagnosis of depression. The scale consists of 21 questions and is used for self-assessment. Its result is an indication, but not a diagnosis. There are 4 possible answer variants. Subsequent answer choices correspond to increased intensity of symptoms, with increasing scores from 0 to 3 ill-being points. Interpretation: 0–13 no depression; 14–19 mild depression; 20–28 moderate depression; 29–63 severe depression [23].

Internet Use Test, developed by Ryszard Poprawa from the Institute of Psychology at the University of Wrocław, shows psychological, social, and health problems caused by Internet use. The total score is the sum of 23 ratings on a scale from 0—‘never’ to 5—‘always’. The minimum raw score is 0 points and the maximum score is 115 points. The higher the score, the stronger the problematic Internet use; 50 points and above reflect strong compulsive Internet use [24].

GAD-7 is a screening tool used to determine feelings associated with generalized anxiety disorder. It serves as a guide to deciding whether a visit to a psychologist or psychiatrist is indicated, but not as a basis for diagnosis. A score of 0, 1, 2, or 3 is given for experiencing “no symptoms at all”, “several days”, “more than half the days”, and “almost every day”, respectively. Scores are summed and presented from 0 to 21. Scores of 5, 10, and 15 represent cutoff points for mild, moderate, and severe anxiety, respectively. Further evaluation is recommended when a score is 10 or higher [25].

Buss and Perry Aggression Questionnaire contains 29 questions to measure aggressive tendencies (both physical and verbal aggression), as well as anger and hostility. The subject marks answers ranging from 1—“extremely uncharacteristic of me” to 5—“extremely characteristic of me”. The questionnaire covers four factors: Physical Aggression (PA), Verbal Aggression (VA), Anger (A), and Hostility (H). The total score is the sum of the scores obtained by the subject on the individual factors [26].

The overall study was conducted in accordance with the standards of the Declaration of Helsinki and was approved by the Bioethics Committee of the Pomeranian Medical University (KB-0012/90/18).

The project was co-financed by the Fund for the Solution of Gambling Problems at the disposal of the Minister of Health, under the agreement concluded between the Minister of Health, represented by the Director of the National Bureau for Drug Prevention, and the Pomeranian Medical University in Szczecin for the period 2 January 2020–31 December 2021.

### 2.4. Statistical Analysis

Statistical analysis was performed using Statistica version 13 software (StatSoft, Krakow, Poland). Continuous variables were characterized by arithmetic means (X), standard deviation (SD), and data range. In the description of qualitative variables, the integer number (n) was presented along with the percentage (%). The distribution of the data was tested using a Shapiro–Wilk test. Due to the non-parametric nature of the variables and the multiple groups, it was decided to use a Kruskal–Wallis test. Kruskal–Wallis tests were used to evaluate the relationship between groups. A post hoc analysis was used to compare multiple groups. Spearman rho correlation analyses and multivariate linear regressions were also performed. In multivariate analysis, the dependent variable was the achieved score in the Beck questionnaire. The results were presented using adjusted odds ratio (OR), beta (β) regression coefficient, and statistical significance (p). The independent variables were the scores obtained from the other questionnaires. The significance level was set at *p* ≤ 0.05.

## 3. Results

The group of male respondents in this study was divided into 3 groups according to the results of the Internet Use Test, namely low, moderate and high problematic Internet use, with 19.4% (*n* = 92) showing low problematic Internet use, 62.0% (*n* = 305) moderate problematic Internet use and 17.8% (*n* = 85) high problematic Internet use. A scale of very high problematic Internet use was shown for 3 (0.8%) of the subjects, so for the purpose of data analysis they were included in the high-use group, resulting in 88 (18.6%) subjects in the high-use group.

The most frequent devices used for Internet access were the “cell phone and tablet” 53% (*n* = 265), followed by computer 29% (*n* = 145) and just the cell phone 18% (*n* = 90). 

Greater problematic Internet use was associated with the duration of Internet studying during the week (*p* = 0.007) and for Internet use for other purposes during both the week and the weekend (*p* < 0.001) (Table 1).

The largest percentage of subjects (62%) showed moderate problematic Internet use, studying online for 15 h on weekdays, using the Internet for other purposes for 12 h during the working week, and 24 h on weekends, which was also shown by the group with high problematic Internet use according to IUT.

A post hoc analysis was performed to find correlations between groups regarding Internet use for other purposes than work or study during weekdays. A statistically significant difference was found between the groups with moderate and high problematic Internet use (*p* = 0.025), and between low and high problematic use (*p* < 0.0001). Analyzing the number of hours used for Internet activity on weekends, there was a correlation between the groups showing moderate and high problematic Internet use (*p* = 0.004) and between those reporting low and high problematic Internet use (*p* < 0.0001).

High problematic Internet use was associated with higher symptoms of anxiety, depression and aggression (verbal and physical) (Table 2).

In the analysis of relationships between IUT problematic Internet use and individual risky behaviors and symptoms in the subjects, statistically significant differences were found between problematic Internet use and anxiety symptoms (GAD-7) (*p* < 0.001), depressive symptoms (Beck Depression Inventory) (*p* < 0.001), anger (*p* = 0.003), and hostility (*p* < 0.001).

In a between-group analysis of statistically significant results, statistically significant differences (*p* < 0.001) for both anxiety symptoms (GAD-7) and Beck Depression Inventory symptoms were found for the three IUT problematic Internet use groups. The level of anger (Buss-Perry questionnaire) also had a statistically significant difference between the low and moderate (*p* = 0.03) and between the low and high problematic Internet use groups (*p* = 0.01). In the dimension of hostility, all groups differed significantly from each other (*p* < 0.001).

The analysis between symptoms according to the Beck Depression Inventory and problematic Internet use shows that 18.6% (n = 93) of low problematic Internet users showed no symptoms of depression. Additionally, 0.4% (n = 2) of people showed a moderate degree of depressive symptoms. Among the moderate problematic Internet users according to the Internet Use Test (IUT), 56.4% (n = 282) showed no depressive symptoms, 5.8% (n = 29) reported mild symptoms, 1.8% (n = 9) moderate symptoms and 0.4% (n = 2) severe symptoms of depression. Among high problematic Internet users, those without depressive symptoms accounted for 9.8% (n = 49), mild depressive symptoms were shown by 4.4% (n = 22) of respondents, moderate symptoms by 1.4% (n = 7) and severe symptoms by 1.0% (n = 4).

Spearman’s rho correlation analysis was also performed. In this analysis, a significant correlation was found between the absence of depressive symptoms in the Beck Depression Inventory and the presence of anxiety according to the GAD-7 scale (*p* = 0.000), and the presence of verbal aggression (*p* = 0.020), anger (*p* = 0.042) and hostility (*p* = 0.000). A significant correlation was also shown in mild value on Beck depressive symptoms scale and verbal aggression (*p* = 0.032) and physical aggression (*p* = 0.033).

Multivariate linear regression shows correlations between problematic Internet use and the results of the used questionnaires. There was a relationship between problematic Internet use and anxiety symptoms in the GAD-7 questionnaire (*p* = 0.000; β = 0.251) and a relationship between problematic Internet use and depressive symptoms (*p* = 0.000; β = 0.269) (Table 3).

Multivariate linear regression shows correlations between depressive symptoms in the Beck Depression Inventory divided into degrees of severity of depressive symptoms and individual questionnaire scores. A significant association was shown between the presence of depressive symptoms according to the Beck Depression Inventory and anxiety symptoms according to the GAD-7 questionnaire (*p* = 0.001, β = 0.350), aggression (*p* = 0.000, β = 1.318), verbal aggression (*p* = 0.002, β = −0.567), physical aggression (*p* = 0.008, β = −0.459) and problematic Internet use (*p* = 0.040, β = 0.197).

A significant correlation was also shown between moderate Beck Depression Inventory symptoms versus anxiety symptoms according to the GAD-7 questionnaire (*p* = 0.000, β = 0.523) and between severe depressive symptoms according to the Beck Inventory versus anxiety symptoms in the GAD-7 questionnaire (*p* = 0.005, β = 0.307) and problematic Internet use (*p* = 0.035, β = 0.204) (Table 4).

## 4. Discussion

This study analyzed the relationship between problematic Internet use and the incidence of aggression, anger, hostility, anxiety, and depression in a group of young men. In addition, it also evaluated whether the degree of depression had an influence on the occurrence of problematic Internet use, verbal aggression, physical aggression, anger, hostility, and anxiety.

Research shows how common a tool the Internet is becoming in an increasingly wide age range of society. Our research shows the age range 18–30 years, known to be the most vulnerable to problematic Internet use and the related dangers [27].

The results of our study show that moderate to high Internet use is associated with more risky behaviors and the display of hostility and aggression, as well as the occurrence of anxiety and depression.

Researchers in Lebanon studying a young population show similar results—higher levels of aggression, depression, impulsiveness, and social anxiety were associated with higher levels of Internet addiction [28]. Researchers in Brazil, based on the Internet Gaming Disorder Scale-Short-Form (IGDS9-SF), found the prevalence of IGD in a group of students at 38.2%, and 18.2% at risk for IGD [29]. The results of others vary—one conducted on a large and representative sample reported prevalence of IGD at less than 5% [30], and another one at 15.7% [31]. Researchers from Portugal investigating the effects of violent games on player behavior in a small group (n = 100) of students found that their emotional state was not affected, but 7.4% did feel more aggressive [32]. Further analysis showed that IGD risk factors were similar to those reported in previous cross-sectional studies, such as insomnia and depressive symptoms [31,33]. Moreover, most participants at risk for IGD presented with severe depressive symptoms [29].

Similar to our study, other researchers found an association between problematic Internet use and the presence of depression and anxiety [33,34]. Interestingly, there are several studies that find links between problematic Internet use and anxiety, depression, and other disorders, showing problematic Internet use largely as the cause of the disorders rather than the consequence [35]. There is a study which shows the presence of depression and social anxiety as predictive factors for Internet addiction [36]. It is increasingly suggested that the relationship between inappropriate Internet use and the emotional problems associated with it may be bidirectional [37,38].

Interesting correlations are observed in this work by analyzing the Beck Depression Inventory symptom severity in relation to individual scales. Most correlations between the GAD-7 scale, the Aggression Questionnaire (including Verbal and Physical Aggression) and the Internet Use Test and the Beck Depression Inventory are visible in mild depression. It should be mentioned that Beck Depression Inventory is a preliminary scale and does not constitute a diagnosis. It informs about the possibility of a problem in a given degree of severity. These tests give us extremely important information that a problem is indeed present, but more detailed investigations are required to understand its nature.

Individuals at risk of addiction spend on average more than 3 h per day in front of a screen [39]. The most commonly used devices for online activities in Poland in 2012 were computers (62%), computer and mobile devices (34.7%), and just mobile devices (3.3%) [28]. In our study, which took place almost a decade later, the Internet was accessed mostly via cell phone and tablet (53%), followed by computers (29%) and just cell phones (18%). Communication, work and online entertainment are now available more using mobile and compact devices, which may represent a greater risk of Internet abuse.

### Limitations

There are many publications focusing on problematic Internet use, but many researchers use multiple survey questionnaires that have different cutoff points that hinder reliable comparisons between populations. In addition, a questionnaire survey is a subjective method where responses can depend on many factors, including, but not limited to, the respondent’s mood at the time. One of the difficulties in identifying problematic Internet use is also the ambiguity and variety of terminologies used. Standardizing the terminologies would make it easier to navigate this sensitive topic. The identification of risk factors and protective factors for Internet Addiction is important for both research and clinical practice. This study indirectly contributes to this identification, showing susceptibility to behavioral disorders, helping in early detection, and opening new avenues of discussion. In addition, our study was conducted in a stationary setting during the COVID-19 pandemic, which increased the difficulty with recruiting respondents.

Despite these limitations, our findings help fill the research gap on problematic Internet use and addiction in the Polish population and provide useful data that can be used for cross-cultural studies.

## 5. Conclusions

The Internet is an everyday tool about which society should have adequate knowledge in order to avoid risky behavior.

Problematic use of the Internet may result in the occurrence of anxiety, anger, hostility or depression. Longer duration of Internet activity was correlated to higher scores on the Internet Use Test degree of problematic Internet use. There was a correlation between the severity of mild depression symptoms and the occurrence of anxiety, verbal and physical aggression, and problematic Internet use.

It is advisable to take preventive measures in the form of promoting healthy lifestyles, such as a balanced diet, adequate sleep duration and quality, daily physical activity, avoiding stress, reduced intake of stimulants, and positive thinking. This will promote fewer hours spent on the Internet and positively affect the health status of young men.

## Figures and Tables

**Table 1 ijerph-19-12153-t001:** Number of hours spent online for various purposes according to the Internet Use Test (IUT).

Parameter	Low Problematic Internet Use According to IUT				Moderate Problematic Internet Use According to IUT				High Problematic Internet Use According to IUT				*p*
	X	Min	Max	SD	X	Min	Max	SD	X	Min	Max	SD	
**Leisure time-weekdays (h)**	4.34	1.00	10.00	2.55	4.60	1.00	16.00	2.75	4.67	1.00	15.00	2.63	0.587
**Leisure time-weekend (h)**	8.52	0.00	24.00	5.19	8.94	0.00	27.00	4.65	9.43	2.00	24.00	5.01	0.495
**Internet use for studying-weekdays (h)**	2.68	0.00	10.00	2.57	3.59	0.00	15.00	3.00	3.63	0.50	10.00	2.56	**0.007**
**Internet use for other purposes-weekdays (h)**	2.44	0.00	10.00	1.89	2.93	0.00	12.00	1.80	3.75	0.00	12.00	2.16	**<0.001**
**Internet use for studying-weekend (h)**	2.19	0.00	10.00	2.20	2.31	0.00	15.00	2.21	2.67	0.00	11.00	2.40	0.345
**Internet use for other purposes-weekend (h)**	3.07	0.00	10.00	2.36	4.38	0.00	24.00	2.57	5.34	3.00	15.00	2.86	**<0.001**

X—mean; min—minimum; max—maximum; SD—standard deviation; *p*—statistical significance. Values in bold are statistically significant values.

**Table 2 ijerph-19-12153-t002:** Results of the questionnaires used in the study vs. IUT problematic Internet use.

Variable		Low Problematic Internet Use According to IUT				Moderate Problematic Internet Use According to IUT				High Problematic Internet Use According to IUT				*p*
		X	Min	Max	SD	X	Min	Max	SD	X	Min	Max	SD	
**GAD-7**		2.53	0.00	12.0	2.60	5.72	0.00	21.00	4.03	8.73	0.00	20.00	4.89	**<0.001**
**Beck Depression Inventory**		2.62	0.00	24.00	4.10	6.97	0.00	46.00	6.07	12.28	0.00	39.00	8.57	**<0.001**
**Buss-Perry Aggression Questionaire**	**Verbal aggression**	14.34	4.00	25.00	4.26	15.17	5.00	33.00	3.92	15.55	7.00	22.00	4.12	0.086
	**Physical aggression**	18.66	8.00	41.00	5.74	18.95	8.00	36.00	5.17	19.62	9.00	38.00	5.09	0.288
	**Anger**	15.61	7.00	28.00	5.01	17.63	2.00	32.00	5.53	18.16	9.00	29.00	4.86	**0.003**
	**Hostility**	17.03	7.00	35.00	5.53	19.64	7.00	37.00	5.42	22.36	8.00	33.00	5.29	**<0.001**

X—mean; min—minimum; max—maximum; SD—standard deviation; *p*—statistical significance. Values in bold are statistically significant values.

**Table 3 ijerph-19-12153-t003:** Multivariate linear regression based on the Internet Use Test.

Questionnaire	OR Cl −95.00%	OR Cl +95.00%	Beta (β)	*p*
**GAD-7 (* score)**	**0.556**	**1.404**	**0.251**	**<0.001**
**Beck Depression Inventory**	**0.391**	**0.946**	**0.269**	**<0.001**
**Agression Questionnaire**	−1.397	0.742	−0.294	0.548
**Verbal aggression (VA)**	−0.880	1.522	0.076	0.600
**Physical aggression (PA)**	−0.828	1.405	0.089	0.612
**Anger (A)**	−1.113	1.037	−0.012	0.945
**Hostility (H)**	−0.290	1.924	0.270	0.148

OR: Adjusted odds ratio; Beta (β)—regression coefficient; *p*—statistical significance. Values in bold are statistically significant values.

**Table 4 ijerph-19-12153-t004:** Multivariate linear regressions in the Beck Depression Inventory.

Efekt	Mild Symptoms of Depression				Moderate Symptoms of Depression				Strong Symptoms of Depression			
	−95.00% CI	+95.00% CI	Beta (β)	*p*	−95.00% CI	+95.00% CI	Beta (β)	*p*	−95.00% CI	+95.00% CI	Beta (β)	*p*
**GAD-7**	**0.241**	**0.850**	**0.350**	**0.001**	**0.633**	**0.939**	**0.523**	**0.000**	**0.172**	**0.902**	**0.307**	**0.005**
**Aggression Questionnaire**	**0.149**	**0.506**	**1.318**	**0.000**	−0.163	0.530	0.454	0.298	−6.675	19.460	9.903	0.332
**Verbal Aggression (VA)**	**−0.871**	**−0.203**	**−0.567**	**0.002**	−0.730	0.091	−0.207	0.127	−19.347	6.723	−3.037	0.337
**Physical Aggression (PA)**	**−0.567**	**−0.086**	**−0.459**	**0.008**	−0.620	0.122	−0.212	0.187	−19.758	6.334	−3.990	0.308
**Anger (A)**	−0.579	0.014	−0.350	0.061	−0.601	0.096	−0.231	0.155	−19.051	7.183	−3.364	0.370
**Hostility (H)**	0.290	0.092	0.151	0.098	−0.198	0.536	0.151	0.366	−19.085	7.116	−3.696	0.365
**IUT**	**0.047**	**2.054**	**0.197**	**0.040**	−0.006	0.116	0.084	0.079	**0.013**	**0.335**	**0.204**	**0.035**

OR: Adjusted odds ratio; Beta (β)—regression coefficient; *p*—statistical significance. Values in bold are statistically significant values.

## Data Availability

The pooled data that support the findings of this study are available from the corresponding author, N.T., upon reasonable request.

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
