# Peer review of "Emotional State of Young Men in Relation to Problematic Internet Use"

_ijerph, 2022, doi:10.3390/ijerph191912153_

Round 1
Reviewer 1 Report
Introduction:
1) On page 2 line 68, the authors make a valuable note about gambling as a prime behavior involved with "problematic internet use". However, statistics on internet pornographic use and illegal activities should also be mentioned here.
Method:
1) On page 4 line 155, please provide more detail on the analyses that were conducted. Variables should be included for greater clarity to the reader.
2) It would have been advantageous to determine if certain types of internet uses were more harmful and health outcomes than others.
Results
1) Page 6 line 201: When reporting the differences between conditions with regards to depressive symptoms the percentages for each condition do not add to 100 percent. I found interpreting this section to be particularly unclear.
2) Without a clearer description of the variables involved (as well as explanation as to whether the regression models were hierarchical) these results are also very difficult to clearly understand.A better explanation in the data analysis section would help greatly.
Discussion:
1) There should be more of a distinct conclusion as to whether this study proves that problematic internet use is a cause of depression, a consequence, or a mix of the two.
2) Another limitation is that analyses did not include "type of internet use" as a measure or covariate. Is it possible that studying for 20 hours a week is perfectly non-consequential, while gambling, gaming, or consuming pornography is actually more closely tied with the adverse emotional outcomes.
Author Response
Thank you for your valuable comments, which will contribute to the improvement of future research. Below are the responses to the individual points of the review.
Introduction:
Point 1. “On page 2 line 68, the authors make a valuable note about gambling as a prime behavior involved with "problematic internet use". However, statistics on internet pornographic use and illegal activities should also be mentioned here.”
Response 1: While writing an article, we didn't come across on internet pornographic use and illegal activities so easily. Although it is an interesting topic, however, our research does not focus on internet pornographic use. The topic is certainly worth exploring in future research, focusing on it in even greater detail.
Method:
Point 2. “Method: On page 4 line 155, please provide more detail on the analyses that were conducted. Variables should be included for greater clarity to the reader.”
Response 2: Data analysis section has been better explained. Page 4 line 155-157 and 160-163 (in green)
Point 3. “It would have been advantageous to determine if certain types of internet uses were more harmful and health outcomes than others.”
Response 3: We fully agree with it, however, it is difficult to verify at this level of research which of the certain types of internet use is more dangerous. In our opinion, a typical gamble with the money factor is the most dangerous because it affects all areas of life, especially at the time of failure. What is the consequence of the impact on our health.
Results:
Point 4: “Page 6 line 201: When reporting the differences between conditions with regards to depressive symptoms the percentages for each condition do not add to 100 percent. I found interpreting this section to be particularly unclear.”
Response 4: Thank you for your valuable observation. We mistakenly suggested a different database. This has been corrected. Page 6 line 208 (in green)
Point 5: “Without a clearer description of the variables involved (as well as explanation as to whether the regression models were hierarchical) these results are also very difficult to clearly understand. A better explanation in the data analysis section would help greatly.”
Response 5: Data analysis section has been better explained. Page 4 line 155-157 and 160-163 (in green)
Discussion:
Point 6: “There should be more of a distinct conclusion as to whether this study proves that problematic internet use is a cause of depression, a consequence, or a mix of the two.”
Response 6: The study group has participants who volunteered. It would be beneficial for the research if the group would be selected. Therefore, it is difficult to define it clearly. However, it is a valuable consideration to take such steps in future research.
Point 7: “Another limitation is that analyses did not include "type of internet use" as a measure or covariate. Is it possible that studying for 20 hours a week is perfectly non-consequential, while gambling, gaming, or consuming pornography is actually more closely tied with the adverse emotional outcomes.”
Response 7: During the research, we were surprised by the COVID pandemic. Learning at that time was mainly in online form through which we observe some effect. Decision-makers who deal with education recommend returning to traditional forms of education in our country in the future. Nevertheless, it should be recognized as a good subject for future research.
Thank you for your consideration of this manuscript.
Reviewer 2 Report
The article is relevant to the mission of the journal. I consider it relevant for several reasons. 1. It is a study that contributes to increasing the field of knowledge in relation to internet use and the associated health problems it may entail 2. It is of vital importance to provide preventive measures in the form of lifestyle.
The topic of the article "Emotional state of young men in relation to problematic Inter-2 net use" is interesting and a timely study, as it is an emerging research problem in relation to the prevention of problems such as anxiety, anger, hostility or depression.
The paper is well structured, facilitating the understanding of the study. The theoretical foundation is based on the research questions, providing current and new literature in relation to the study problem and the objectives set out.
Objective: The research problem and the objective of the study are well defined.
Method: The type of design used in the study is not explicitly specified. Therefore, this evaluator considers that it should be provided.
The research phases are presented in a clear and structured way and the research questions are answered in a clear and detailed manner.
Results: This evaluator considers that the results shown in terms of the study problem are relevant.
The limitations are not specified in the study, hence the authors should establish a section for this purpose. Likewise, the authors should expand more on the conclusions, especially by detailing what kind of lifestyles are recommended as a form of prevention.
In short, I believe that this is a good work that will contribute to the advancement of knowledge in relation to certain health problems and the use of the Internet.
Author Response
Thank you for your valuable comments, which will contribute to the improvement of future research. Below are the responses to the individual points of the review.
Point 1. “Method: The type of design used in the study is not explicitly specified. Therefore, this evaluator considers that it should be provided.”
Response 1: In the method section there is information about the type of design of the study. Page 2 line 96. (in yellow)
Point 2. “The limitations are not specified in the study, hence the authors should establish a section for this purpose.”
Response 2: For clarity of the study, we marked the section “Limitations”. Page 9 line 311. ( in yellow)
Point 3. “Likewise, the authors should expand more on the conclusions, especially by detailing what kind of lifestyles are recommended as a form of prevention.”
Response 3: The conclusions have been expanded. Page 9 line 336-337 (in yellow)
Thank you for your consideration of this manuscript.